# VERIFICATION LIMITS CODE LLM TRAINING

## ABSTRACT

Large language models for code generation increasingly rely on synthetic data, where both problem solutions and verification tests are generated by models. While this enables scalable data creation, it introduces a previously unexplored bottleneck: the verification ceiling, in which the quality and diversity of training data are fundamentally constrained by the capabilities of synthetic verifiers. In this work, we systematically study how verification design and strategies influence model performance. We investigate (i) **what** we verify by analyzing the impact of test complexity and quantity: richer test suites improve code generation capabilities (on average +3 pass@1), while quantity alone yields diminishing returns, (ii) **how** we verify by exploring relaxed pass thresholds: rigid 100% pass criteria can be overly restrictive. By allowing for relaxed thresholds or incorporating LLM-based soft verification, we can recover valuable training data, leading to a 2–4 point improvement in pass@1 performance. However, this benefit is contingent upon the strength and diversity of the test cases used, and (iii) **why** verification remains necessary through controlled comparisons of formally correct versus incorrect solutions and human evaluation: retaining diverse correct solutions per problem yields consistent generalization gains. Our results show that Verification as currently practiced is too rigid, filtering out valuable diversity. But it cannot be discarded, only recalibrated. By combining calibrated verification with diverse, challenging problem–solution pairs, we outline a path to break the verification ceiling and unlock stronger code generation models.

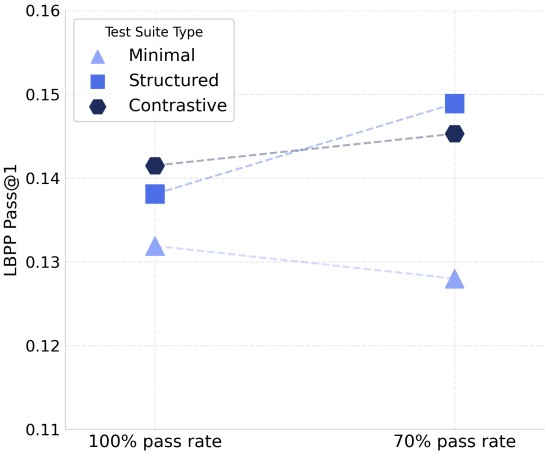

Figure 1: Interaction between test suite complexity design and pass rate thresholds ($\tau = \{0.7, 1\}$) on LBPP performance. Relaxing thresholds under Minimal test suites degrades performance as weak tests admit noisy solutions. In contrast, Structured and Contrastive suites benefit from relaxed thresholds, with Structured yielding the largest gains. Richer test suites provide a higher-resolution verification signal, enabling soft filtering that retains diverse, non-canonical solutions while maintaining some level of correctness.

## 1 INTRODUCTION

The use of synthetic data is becoming a commonplace practice to obtain large-scale training data for LLMs (Chen et al., 2024; Grattafiori et al., 2024; Gunter et al., 2024; Yang et al., 2025; Bercovich et al., 2025). A critical component of this pipeline is verification: filtering model outputs to retain only those deemed correct. Verification prevents the compounding of model errors often discussed under the umbrella of "model collapse" (Zhu et al., 2025) and enables high-quality data curation without extensive human supervision. This is especially powerful in domains with verifiable reward

signals, such as code, where correctness can be automatically tested. In practice, common pipelines (Wei et al., 2024a) not only generate candidate completions synthetically but also synthesize verification: code solutions must pass tests generated by a model, and only passing solutions are retained for training. While this strategy has clear appeal, its effectiveness is not guaranteed. For example, Guha et al. (2025) compared datasets filtered by LLM-generated unit tests against unfiltered counterparts and found no improvement from verification-based filtering, suggesting that naive or rigid applications of synthetic verification may fail to capture the intended quality signal.

These observations point to a deeper and underexplored dependency: The quality of the training data becomes fundamentally constrained by the capabilities of the synthetic verification system. When both solutions and their validation are model-generated, we risk creating a closed loop in which only solutions recognizable to the verifier survive, excluding potentially correct, diverse, or complex implementations that exceed its competence. We refer to this bottleneck as the **verification ceiling problem**. In the context of code, this is particularly concerning, as prior works show that model performance is sensitive to the breadth and quality of supervision (Zelikman et al., 2022; Wang et al., 2023). To study this systematically, we ground our analysis in a suite of standard code generation benchmarks spanning four programming languages, where evaluation is precise and the effects of verification choices can be measured directly. This work asks three core questions:

**RQ1.** *What to verify?* How do the complexity and quantity of test suites used in verification affect the quality of training data? We find that both complexity and number of tests significantly shape which solutions are retained and how effectively the model learns: Increasing test complexity improves downstream model performance by average of +3 pass@1 across comprehensive set of benchmarks. However, simply adding more tests without increasing their sophistication has diminishing or even negative returns.

**RQ2.** *How to verify?* Can alternative strategies, such as relaxing strict pass thresholds or using LLM-as-a-judge signals, outperform brittle criteria like requiring 100% test pass rates? We find that moderately relaxing rigid 100% pass thresholds or incorporating LLM-based soft verification can recover useful solutions that would otherwise be discarded, boosting downstream performance by +2–4 points pass@1. Interestingly, with pass threshold relaxation, this improvement is not universal and only occurs when the verification signal is sufficiently rich with more complex unit tests (Figure 1). This highlights a subtle but important dynamic: soft filtering can enhance learning, but only when the underlying verification is strong enough to provide a high-resolution correctness signal.

**RQ3.** *Why verify at all?* If strict verification can hurt and soft verification already recovers useful data, then what role does verification truly play? To probe this, we conduct a human annotation study of synthetic unit tests, revealing substantial blind spots in their correctness and completeness.

Our results show that verification is essential, yet its value is contingent on execution: overly rigid enforcement can discard valuable solutions, while ignoring verification altogether allows low-quality data to dominate. The true bottleneck is not correctness itself, but the brittle, narrowly defined methods we currently use to enforce it, highlighting the need for a calibrated, nuanced approach. We argue for **calibrated verification** where correctness filtering is neither too lenient nor too strict and for diverse, challenging samples that promote generalization without overfitting to verifier-specific patterns. Our work provides actionable insights for building more effective synthetic data pipelines and lays the groundwork for breaking through the current verification ceiling.

## 2 FORMALIZING THE VERIFICATION CEILING PROBLEM

Here we formalize the code generation training pipeline to precisely characterize the verification ceiling problem. Consider a training dataset $\mathcal{D} = \{(p_i, s_i)\}_{i=1}^{N}$ where each sample consists of a programming problem $p_i$ and a corresponding solution $s_i$. In synthetic data generation, we construct this dataset through a verification-filtered sampling process.

For each problem $p_k$, we generate a set of candidate solutions $\mathcal{S}_k = \{s_k^{(1)}, s_k^{(2)}, \ldots, s_k^{(m)}\}$ using one or more teacher models. Each solution is evaluated against a verification system $V_k$ consisting of a test suite $T_k = \{t_k^{(1)}, t_k^{(2)}, \ldots, t_k^{(l)}\}$ where each $t_k^{(i)}$ is generated synthetically and independently.

The verification function $V_k(s; T_k) \in [0, 1]$ measures the fraction of tests passed, where traditional approaches use a strict threshold: $V_k(s; T_k) = 1$ if and only if solution $s$ passes all tests in $T_k$.

The training dataset is then constructed as:

$$\mathcal{D} = \{(p_k, s) : s \in \mathcal{S}_k, V_k(s; T_k) \geq \tau, \forall k\} \tag{1}$$

where $\tau$ is typically set to 1.0. This formulation reveals the **verification ceiling**: the quality of training data $\mathcal{D}$ is fundamentally bounded by the coverage and correctness of $T_k$ and also the strictness and sophistication of $V_k$. As models and data scale, the verification ceiling constrains the attainable quality and diversity of training solutions, potentially capping downstream model performance gains. Consequently, overcoming this ceiling requires both improving verification methods and rethinking how we leverage diverse signals of solution correctness beyond strict unit test pass rates. In the following sections, we systematically investigate these aspects through controlled experiments, aiming to characterize and ultimately alleviate the limitations imposed by the verification ceiling.

## 3 EXPERIMENT SETUP

**Training Data.** In this paper we investigate how the base model acquires code generation capabilities when finetuned on selectively curated training data. In each experiment, we finetune the base model with datapoints structured as prompts and completion pairs. In particular, the model is trained with a data mixture consisting of:

*1. Synthetic code samples.* Building on the approach introduced in Wei et al. (2024b), we curate novel coding problems from a pool of coding concepts, followed by generating unit tests for self-verification. For each problem, we generate both unit tests and corresponding code solutions using strong teacher models. Optionally, hard problems are allowed to have multiple attempts given the execution feedback from the previous failed generation. Unless specified otherwise, we use Deepseek-v3 (DeepSeek-AI et al., 2025) for all steps of this pipeline and fix the synthetic training set size to 70K per experimental setup.

*2. High-quality publicly available code samples.* We include a fixed amount (70K) of high-quality competitive programming style code data in all experiments. The code mixture is sampled from the following sources: APPS (Hendrycks et al., 2021), CodeContests (Li et al., 2022), TACO (Li et al., 2023), XCodeEval (Khan et al., 2023).

*3. Non-code instruction following samples.* All training mixtures include a 5% portion of non-code samples drawn from general instruction-following tasks, ensuring the model retains capability beyond code generation. This subset consists of reasoning tasks such as math problems, safety related data, as well as general purpose tasks such as translation and summarization.

**Model.** We use the Command 7B base model (Cohere et al., 2025).All trainings are conducted using three random seeds, with reported results representing averaged performance across these runs to enhance the statistical significance and reliability of the observed improvements.

**Evaluation setup.** Code generation benchmarks evaluate models on tasks that require providing code implementation given a programming instruction. We evaluate each finetuned model for code generation performance ( 1-shot pass@1) on HumanEval (Zheng et al., 2024), BigCodeBench (Zhuo et al., 2024), LBPP (Matton et al., 2024), LiveCodeBench (Jain et al., 2024), MBPPPlus (Lyu et al., 2024), and McEval (Chai et al., 2024). See Table 1 for more details on each evaluation set. For aggregated results, we report a weighted average that accounts for the varying dataset sizes across benchmarks and programming languages.

## 4 CHOOSING WHAT TO VERIFY

The verification ceiling manifests most critically in the design and implementation of *what* we choose to verify: the set of $T_k$ unit tests for each code sample. In this section, we examine key factors in test suite generation and analyze how they affect LLM performance.

### 4.1 UNIT TEST COMPLEXITY

We first investigate how using more complex unit-tests affects the training distribution of synthetic generated code data. The motivation here is straightforward: if simple tests only capture surface-level correctness, more complex tests may probe deeper semantic understanding and lead to more

robust filtering. In this setting, we aim to raise the verification ceiling by making each test a stronger signal of correctness.

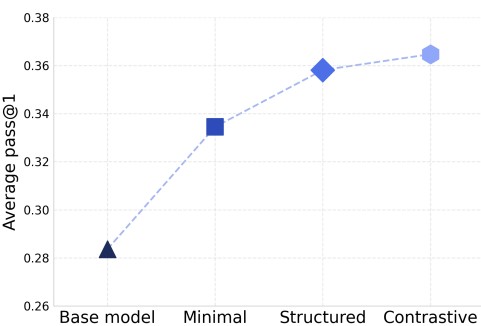

Figure 2: Average pass@1 across all evaluation benchmarks for models trained with different test suite complexities. Base refers to the pretrained model without SFT, while Minimal, Structured, and Contrastive correspond to progressively more sophisticated test suites prompting methods. Structured prompting method yields the largest improvement over Minimal (+3 pass@1), and Contrastive provides further gains (+1 pass@1).

To investigate whether more complex test cases can improve downstream model performance, we experiment with a series of increasingly sophisticated prompting strategies and enriched context to generate synthetic unit tests. We define three variants of prompt-based test generation, each reflecting a different level of context and target complexity (See Appendix A for more details):

**(1) Minimal Prompting**: Given only the problem description $p_k$, we prompt the model to generate a set of unit tests $T_k$ and also include tests for non-trivial inputs. The goal is to encourage the model to go beyond simple or typical inputs and explore a broader range of functional behavior. **(2) Structured Prompting**: Given a problem description $p_k$, a candidate solution $s_k$, and a set of minimal unit tests $T_k$, we prompt the model to generate a new set of tests that are more complete specifically target edge cases and under-tested logic. This approach encourages more targeted verification and improved code coverage. **(3) Contrastive Prompting**: Given the problem description $p_k$, multiple candidate solutions $\mathcal{S}_k$, and an initial test suite $T_k$, we ask the model to generate new tests such that at least one of the solutions fails each test. This encourages the generation of adversarial and high-coverage examples that explicitly differentiate between correct and incorrect behavior.

For each approach, all problems are considered in the candidate pool and solutions are evaluated solely against the unit tests specific to the assigned complexity level. We validate the increase of test suite complexity resulting from interventions by using a separate LLM-as-a-judge (see Appendix A). To compare the impact of test suite complexity, the tests are used to filter synthetic code solutions and select equal number of training samples (70K) per experiment: a candidate is retained only if it passes all the generated tests ($\tau = 1$). Our findings in Figure 2 show a clear benefit of increasing test complexity. We observe that increasing the sophistication of test prompts yields measurable improvements in downstream performance, especially from Minimal to Structured unit tests. Moving from Minimal to Structured yields a +3 pass@1 gain, making Structured a strong setting for filtering brittle or underspecified solutions that would otherwise pass basic checks. Contrastive unit tests provide further improvements over Structured (+1 pass@1) and deliver the highest overall performance, representing a +7 points absolute gain compared to the base model. While the marginal improvement beyond Structured is smaller, this suggests that contrastive tests complement structured ones by capturing a different class of subtle errors and promoting more robust solution quality. These results show that while increasing test complexity improves filtering precision, the biggest gains come from moving beyond minimal verification. Both Structured and Contrastive test suites highlight that better verification design can directly translate into stronger downstream models.

## 4.2 UNIT TEST QUANTITY

Next we study raising the verification ceiling by asking: how many unit tests are sufficient to reliably distinguish between correct and incorrect code solutions without overly filtering out valuable instances? We conduct a controlled experiment where we generate candidate code solutions and validate them using different number of synthetic unit tests per sample: $\{1, 2, 3, 4\}$. In all cases, test generation pipeline is held constant and only the number of tests used for filtering varies. We then sample 70K synthetic samples with $\tau = 1$ under each filtering condition.

Interestingly, we observe a non-monotonic trend in Figure 3 (left). Moving from one to two unit tests leads to a clear improvement of +3 pass@1 points in downstream accuracy, indicating that the

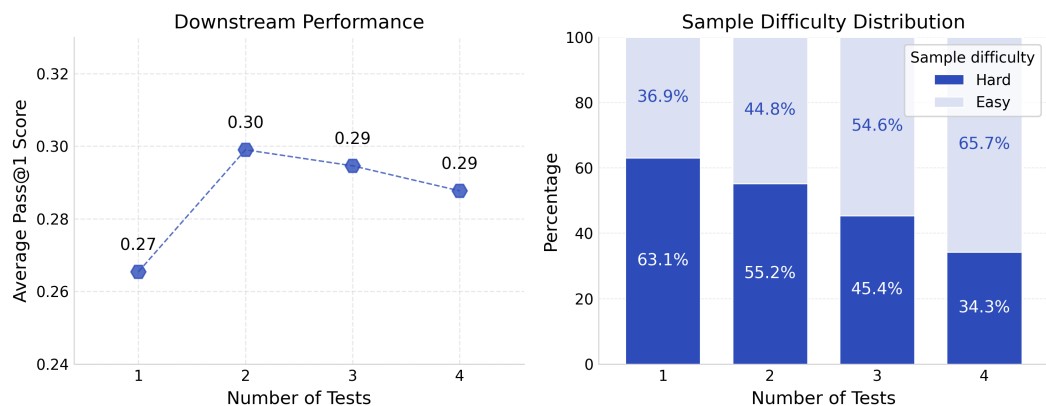

Figure 3: Effect of increasing the number of required test passes on model performance and training data composition. **Left:** Average pass@1 across benchmarks shows a non-monotonic trend: moving from one to two tests improves performance (+3 points), but stricter filtering beyond two tests degrades accuracy. **Right:** Distribution of problem difficulty under different verification regimes, annotated with Command-A. Stricter filtering disproportionately removes harder problems indicating that overzealous verification selects for simplicity rather than robustness.

additional verification step effectively filters out spurious or brittle solutions without being overly restrictive. However, beyond two tests, performance begins to degrade and the model trained on the more strictly filtered data underperforms relative to the previous baseline. To characterize how test-based filtering alters the composition of our training corpus, we perform an annotation of problem difficulty across verification regimes. Given the problem definition and code solution $(p_i, s_i)$ we prompt Command A (Cohere et al., 2025) to rate the difficulty level of the problem and assign a score between 1 and 5. We then group the samples into three difficulty class: easy (scores 1 and 2), medium (score 3), and hard (scores 4 and 5). In Figure 3 (right) we observe that increasing the number of required test passes disproportionately removes more difficult samples. In effect, we are not just filtering for correctness, we are filtering for simplicity and verifier alignment.

These results underscore a broader insight: increasing the number of unit tests is not necessarily a reliable strategy for improving data quality. While stricter verification does reduce incorrect solutions, it also disproportionately filters out hard coding problems and non-trivial, partially correct implementations that fail due to weaknesses in the test generation process itself. This leads to a subtle but important shift in the training distribution: the surviving examples tend to be simpler, biasing the model toward easier patterns and reducing its exposure to complex or creative code.

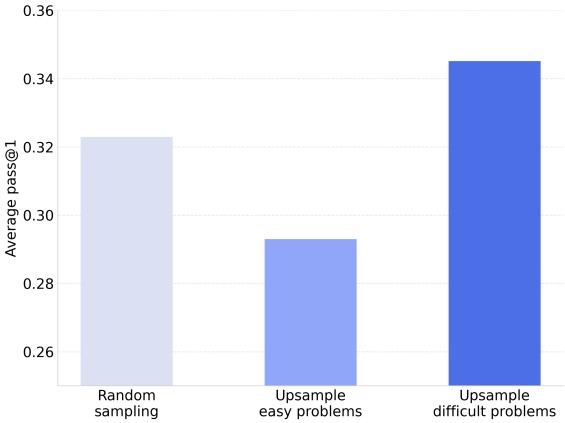

Figure 4: Impact of problem difficulty on training effectiveness. Models trained with datasets enriched with harder problems (40% hard, 40% medium, 20% easy) consistently outperform those skewed toward easier problems (20% hard, 40% medium, 40% easy).

To disentangle the effects of test-based filtering from the intrinsic value of training on more difficult problems, we conduct an additional experiment where we directly control the difficulty of the problems included in the training set. Instead of varying the number of unit tests for verification, we

sample training examples according to their difficulty score. In all settings, 40% of the data consists of medium-difficulty problems. The remaining 60% is divided differently depending on the upsampling strategy: in the easy-upsample setting, it includes 40% easy and 20% hard problems, whereas in the hard-upsample setting, it includes 40% hard and 20% easy problems. To make sure we have sufficient data points with the right difficulty balance for each training, we relax the verification criteria to $\tau = 0.6$. We then compare models trained on datasets enriched with harder problems versus those skewed toward easier ones, keeping dataset size and other factors constant (Figure 4). Remarkably, we find that models trained on more difficult problems consistently outperform those trained on easier ones by 6 points on average pass@1 across all evaluations, even when the associated solutions are not strictly verified. This finding suggests that the objective should not be maximizing the number of tests passed, but rather designing verification systems that are calibrated to retain valuable diversity while enforcing correctness.

## 5 Choosing How to Verify

The previous section explored verification through increasingly strict unit test regimes, showing that both test quantity and complexity shape which solutions are accepted into the training distribution. However, these approaches implicitly rely on a binary notion of correctness: solutions must pass all tests to be considered valid ($\tau = 1$). In this section, we examine whether relaxing this rigid criterion can lead to better learning outcomes.

We first investigate partial-pass filtering, where we accept solutions that pass a high (but not perfect) fraction of the test suite. This approach assumes that a solution passing tests partially may still contain useful signal, especially if failures are due to narrow test cases or minor implementation mismatches. We then explore a more flexible alternative by replacing unit-test-based filtering with LLM-based judgments, where a language model directly assesses the quality of candidate solutions. This shift opens the door to verification criteria that incorporate fluency, intent alignment, or plausibility, dimensions that are difficult to capture with rigid programmatic tests alone. Together, these strategies reflect a move from strict correctness filtering toward a broader notion of usefulness in training data, allowing for richer and more diverse solution spaces.

### 5.1 Relaxing Unit Test Verification criteria

The formalization of verification in Section 2 shows that standard synthetic training pipelines rely on a strict correctness threshold: a solution is retained only if it passes 100% of its associated tests ($\tau = 1.0$). This threshold enforces high precision but also constrains recall. Many partially correct, semantically meaningful, or pedagogically valuable solutions are excluded simply because they fail a single test. To investigate the potential benefits of loosening this constraint, we experiment with relaxing the pass-rate threshold $\tau$ during data selection.

For each problem $p_k$, we generate candidate solutions $\mathcal{S}_k$ and filter them based on a relaxed criterion $V_k(s; T_k) \geq \tau$, where $\tau = \{0.1, 0.2, 0.4, 0.6, 0.8, 1\}$. We then train models on the filtered datasets at different $\tau$ levels and evaluate their performance on a held-out benchmark. Note that all experiments use the same training data size, with 70K synthetic code samples included in each mixture. In Figure 5 we observe a surprising trend: enforcing full test suite pass ($\tau = 1$) does not yield the best downstream performance. In fact, moderately relaxed thresholds, particularly in the 0.6 to 0.8 range, tend to produce better results across most programming languages.

Next, we examine how the type of test generation prompting method interacts with the pass rate threshold to influence training outcomes. We conduct this experiment under three different setups: Minimal, structured, and Contrastive Test suite generation. Different test construction strategies shape the nature of the verification signal and what types of solutions are admitted into the training set at a given $\tau$. Our results in Figure 1 on the difficult LBPP benchmark show that the impact of relaxing the pass rate is highly dependent on the nature of the test suite. Under Minimal test suites, lowering $\tau$ leads to a drop in downstream performance by 0.5 point. Because the tests are weak proxies for correctness, relaxing the threshold merely allows through more incorrect or degenerate solutions, introducing noise into the training data. However, with structured and contrastive test suites, relaxing $\tau$ to 0.7 leads to consistent improvements in downstream performance with the structured setting exhibiting the largest gains (+1 pass@1). The richer tests are strong enough to

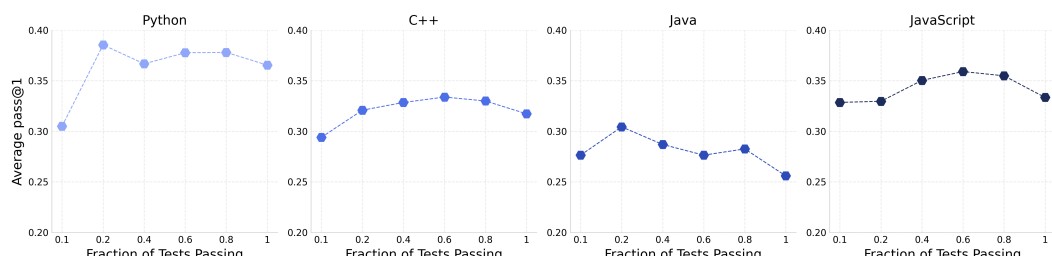

Figure 5: Effect of varying verification thresholds ($\tau$) on downstream performance across programming languages. Strict enforcement ($\tau = 1$) consistently underperforms, while moderately relaxed thresholds (0.6–0.8) yield the best results.

filter out clearly incorrect solutions, even under relaxed thresholds, while allowing more diverse and non-canonical implementations that would otherwise be discarded under the strict $\tau = 1.0$ regime. In essence, complex tests provide a high-resolution signal that supports soft filtering, whereas simple tests require binary enforcement to avoid noise.

These findings reinforce our central argument: optimizing for correctness alone, especially under limited or weak verification, is insufficient. Structured, expressive test suites allow for graded notions of correctness, enabling the inclusion of useful training signals even from partially passing solutions. This represents a practical path for lifting the verification ceiling by improving the alignment between verification granularity and selection criteria.

## 5.2 BEYOND UNIT TESTS: DIRECT LLM VERIFICATION

While unit tests provide a concrete verification signal, they are inherently limited by their coverage and sensitivity to superficial variations in code. To address these limitations, we explore an alternative verification approach where LLMs serve as plausibility and correctness evaluators. Formally, given a problem $p_k$ and a candidate solution $s \in \mathcal{S}_k$, we replace the traditional unit-test-based verification function $V_k(s; T_k)$ with an LLM-based verifier $L_k(s; M) \in [0, 1]$ that scores the solution based on its plausibility, idiomatic usage, and likely correctness using language model $M$. Under this setup, the training dataset is constructed as:

$$\mathcal{D} = \{(p_k, s) : s \in \mathcal{S}_k, L_k(s; M) \geq \tau, \forall k\} \tag{2}$$

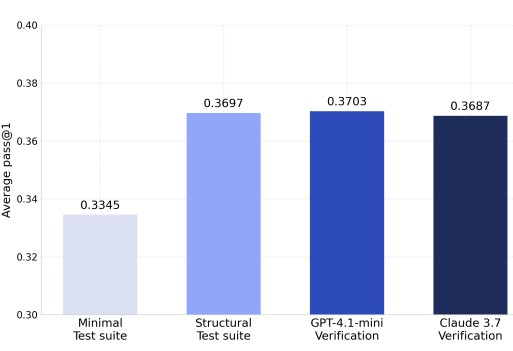

Figure 6: Comparison of unit test–based and LLM-based verification strategies. Structured test suites provide the strongest gains over Minimal verification, while LLM-based filtering performs on par with Structured verification and consistently outperforms Minimal. Overlap analysis reveals that LLM judges select partially overlapping but distinct subsets of data (Claude-3.7 vs. GPT-4.1-mini: 49%; Structured Test Suite vs. Claude-3.7: 32.6%; Structured Test Suite vs. GPT-4.1-mini: 32.9%), suggesting that LLMs capture complementary signals beyond traditional test-based criteria.

where $\tau$ is a threshold on the LLM judgment score. This formulation allows us to evaluate whether using LLM-based verification can recover high-quality solutions that would otherwise be discarded by strict or brittle unit-test criteria, and how it interacts with the verification ceiling. We experiment with two LLMs as verifiers tasked with scoring candidate solutions along axes such as correctness,

code quality, and alignment with common programming practices: (1) **GPT-4.1-mini:** We use GPT-4.1-mini at temperature 0 to select one response from a set of attempts corresponding to every code problem that the model deems correct. (2) **Claude-3.7-sonnet:** For the same process, we then consider a different model as the judge namely Claude-3.7-sonnet, which is a stronger reasoning model, at temperature 0 to again select one response from a set of attempts corresponding to every code problem that the judge deems correct.

The system prompt used to choose the solution is included in Appendix D. We then finetune our model on the solutions picked by LLMs as the judge and compare it with the model finetuned on solutions picked by unit-tests as the verifier as described above. Despite the lack of formal test execution, both LLM-based filters produce training data that leads to strong downstream performance, comparable to or exceeding that of unit test–based filtering in some settings (Figure 6). This suggests that well-instructed LLMs can serve as effective and flexible judges of solution quality, offering a softer and potentially more generalizable alternative to rigid test-based criteria.

## 6 WHY WE CANNOT SKIP VERIFICATION

Up to this point, our experiments have shown that strict reliance on synthetic unit tests can inadvertently bias the training distribution toward simpler problems and discard valuable solutions. In this section we study why abandoning verification altogether is not an option.

To probe the reliability of synthetic unit tests and their limitations as a stand-in for verification we conduct a human study with code experts at two levels of experience: junior and senior. We sample 300 python samples from our synthetic data generation pipeline where 100 samples are labeled as incorrect and 200 samples are labeled as correct by Minimal unit tests. Junior coding experts are tasked with labeling the correctness and completeness of unit tests given the instruction. Subsequently, they are asked to provide additional unit tests, specifically targeting edge cases and proper error handling with only problem description and synthetic unit tests provided. Finally, two senior coding experts are asked to further review the unit tests from junior coding experts where they can either fix or provide more test cases. For each sample, the solution is verified against synthetic unit tests, junior-expert-written unit tests and senior-expert-written unit tests. We use Cohen's Kappa ($\kappa$) to quantify the agreement between different types of unit tests.

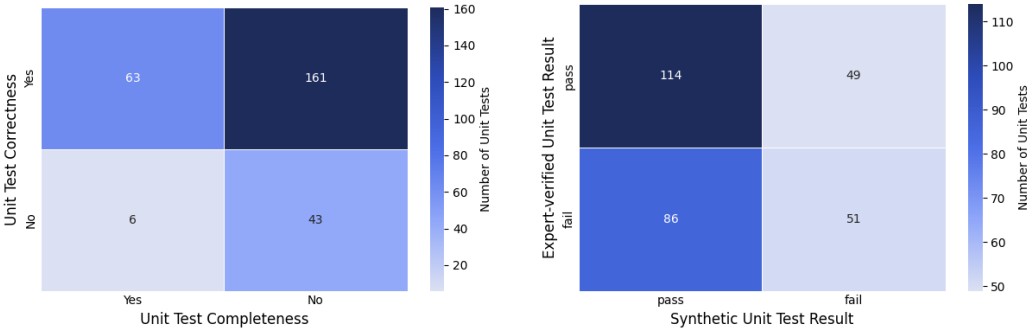

Figure 7: **Left:** Human evaluation of synthetic unit tests. 53.6% of unit tests are correct but incomplet suggests that false positives primarily arise from coverage gaps rather than incorrect test logic. Note that 27 samples were labeled "unsure" are excluded from this plot. **Right:** Agreement count data of synthetic unit tests labels and expert-verified unit tests labels. $\kappa = 0.07$.

Figure 7 (left) shows that only 21% of unit tests are correct and complete and 53.6% of unit tests are correct but incomplete. False positive samples are more likely to be coming from the fact that the synthetic unit tests do not have full coverage of the problem, rather than the synthetic unit tests are wrong. The correct rate of 76.3% suggests that true negative samples are effectively removed by the verification process. Figure 7 (right) shows weak agreement ($\kappa = 0.07$) between synthetic unit tests and expert-written unit tests. Surprisingly, 49% of the samples that fail synthetic tests pass tests written by junior code experts, which indicates that a substantial amount of useful and correct generations for training are potentially discarded. We also report the human unit tests agreement between

junior and expert coding annotators to be $\kappa = 0.77$. To address the confounding factor of synthetic unit tests correctness, we report the agreement of the subset with verified synthetic unit tests correctness. However, the agreement for the synthetic unit tests correctness verified subset is also weak ($\kappa = 0.09$). A Figure 9 in the Appendix presents the overlap statistics between sample execution results with synthetic unit tests and expert unit tests, and their relationship with the correctness of synthetic unit tests. Together, these findings underscore a key point: synthetic unit tests provide a scalable but shallow verification signal, one that is prone to coverage failures. Human evaluation reveals that many discarded solutions are in fact valid, pointing to the need for stronger unit test generation methods, multi-turn test refinement, or hybrid verification strategies that combine automated filters with more capable LLM or human-in-the-loop judgments. Thus, while synthetic unit tests alone are not sufficient, some form of verification—human, LLM-based, or hybrid—remains indispensable to avoid undermining the quality of the training distribution.

## 7 RELATED WORK

The use of synthetic data to train and evaluate code generation models has gained traction as a scalable alternative to curated datasets. Frameworks like Case2Code (Shao et al., 2025) demonstrate that synthesizing diverse code samples via LLMs can improve inductive reasoning and code understanding. Similarly, Nadăş et al. (2025) surveys LLM-driven synthetic data pipelines, highlighting techniques such as prompt-based generation and reinforcement learning with execution feedback. Pipelines for synthetic code generation rely on some form of verification of the generated code as a proxy for its quality (Luo et al., 2025; Chen et al., 2021). The most common method by far relies on unit tests to validate functional correctness, as this also mirrors what is done in popular coding benchmarks like HumanEval (Balloccu et al., 2024), MBPP (Lyu et al., 2024) or LBPP (Matton et al., 2024). However, this approach has potentially several flaws: Unit tests may fail to cover edge cases, overfit to specific implementations, or discard valid solutions that fail partial tests. Moreover, generating comprehensive test suites is labor-intensive and brittle for complex problems (Lahiri et al., 2023), which is only partially alleviated by synthetic methods for unit-test generation (Wang et al., 2025). Moreover, problem difficulty and diversity have a huge impact on the final model performance. Tambon et al. (2025) highlights that model performance degrades significantly as problem complexity increases, suggesting that training data must better reflect this diversity. Chen et al. (2024) also argue that diverse synthetic samples improve generalization, even for smaller models.

## 8 CONCLUSION

In this work, we investigated the **verification ceiling problem** in synthetic code generation, where the quality and diversity of training data are fundamentally constrained by the capabilities of the verification system. Through systematic experiments we demonstrated that test suite complexity and design significantly influence downstream model performance, with more sophisticated tests improving model quality. We showed that alternative verification strategies, such as relaxing strict pass thresholds or using LLM-based judgments, can recover valuable solutions and improve training outcomes. Finally, we establish that verification signals, even if imperfect, are crucial and optimizing how they are applied opens exciting opportunities for stronger, more generalizable code generation models. We advocate for designing verification mechanisms that balance correctness with diversity and challenge, enabling richer training distributions that push beyond current verification ceilings. Our work provides actionable insights for constructing more effective synthetic data pipelines and advancing the capabilities of code generation models.

## 9 LIMITATIONS

In this work, we focus exclusively on supervised fine-tuning using filtered data. While LLM-based verification and relaxed pass thresholds already improve training distributions, we have yet to explore reinforcement learning from verification signals or reward-based optimization strategies. Integrating verification more directly into the learning objective is a promising direction that could unlock even greater performance gains.

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

# A VALIDATING PROGRESSIVE TEST SUITE COMPLEXITY

To validate that our test suite interventions genuinely increase verification difficulty, we conduct a series of head-to-head comparisons using a different LLM as the judge (Command-R+). Specifically, we evaluate whether each successive round of test construction produces test suites that are increasingly challenging. We use 100 coding problems (25 each in Python, Java, C++, and JavaScript), and for each problem, present the LLM with two alternative test suites. We use the following prompt to choose the more challenging test suite between two test suites.

---

USER: Given the following code problem and two set of test suites associated with it, your goal is to choose the more challenging test suite.
Code Problem:

{code_problem}

Test Suite A:

{test_A}

Test Suite B:

{test_B}

A test suite is considered more challenging according to the following criteria:
- Comprehensive Coverage: A challenging test suite should aim to cover a wide range of scenarios, including edge cases, boundary conditions, and various input combinations. It should not leave any critical functionality untested.
- Diverse Test Cases: The test cases should be diverse and vary in complexity. This includes testing different input types, sizes, and formats, as well as exploring various execution paths and code branches.
- Real-World Scenarios: Simulate real-world usage as closely as possible. Include test cases that mimic actual user behavior, common use cases, and potential error scenarios that users might encounter.
- Negative Testing: In addition to testing valid inputs, a robust test suite should also focus on negative testing. This involves providing invalid, unexpected, or erroneous inputs to ensure the system handles them gracefully and provides appropriate error messages or fallback mechanisms.
- Performance and Stress Testing: Consider including tests that evaluate the system's performance under different load conditions. This can help identify bottlenecks, memory leaks, or other performance-related issues.
- Security Testing: If applicable, include tests that assess the system's security measures. This might involve attempting common attack vectors, such as SQL injection, cross-site scripting (XSS), or authentication bypass.
OUTPUT FORMAT: Your output should be a JSON with the following two keys:

```
{
    "assessment_explanation": "<explanation>",
    "test": "<A or B depending on the chosen test-suite>"
}
```

---

In the first comparison, we contrast minimal test suites with structured ones and as shown in Table 2, the LLM selects the structured suite as more challenging in 92 out of 100 cases. In the second comparison, we evaluate whether the contrastive test suites are more challenging than the structured suites. As Table 2 shows, the contrastive suites are deemed more challenging in 64 out of 100 cases. Together, these results confirm that our test generation pipeline yields progressively more challenging verification setups, aligning with our intended design.

| Dataset | Python | C++ | Java | JS | Additional Information |
|---|---|---|---|---|---|
| BigCodeBench (Zhuo et al., 2024) | 1140 | – | – | – | Focus on software-engineering problems (instead of competitive programming), involving diverse function calling and complex instruction following. |
| HumanEval (Zheng et al., 2024) | 164 | 164 | 164 | 164 | Collection of parallel code generation examples assessing language comprehension, algorithms, and simple mathematics. |
| LBPP-v2.1 (Matton et al., 2024) | 161 | 161 | 158 | 153 | Parallel and human-annotated designed to be more difficult than similar datasets |
| LiveCodeBench-v5 (Jain et al., 2024) | 873 | – | – | – | A newer (leakage-free) set of competitive programming questions. |
| McEval (Chai et al., 2024) | 50 | 50 | 53 | 50 | Human-written problems and solutions. |
| MBPPPlus (Lyu et al., 2024) | 378 | – | – | – | A filtered version of MBPP. For evaluation we used the non-plus version (original test sets). |

Table 1: Details of evaluation set for code generation task.

| Comparison | Suite A Preferred | Suite B Preferred | Total |
|---|---|---|---|
| Minimal vs Structured | 8 | 92 | 100 |
| Structured vs Contrastive | 36 | 64 | 100 |

Table 2: LLM-based comparison of test suite difficulty. Each row reports the number of times a given test suite was preferred as more challenging in a head-to-head comparison by Command-A (temperature 0).

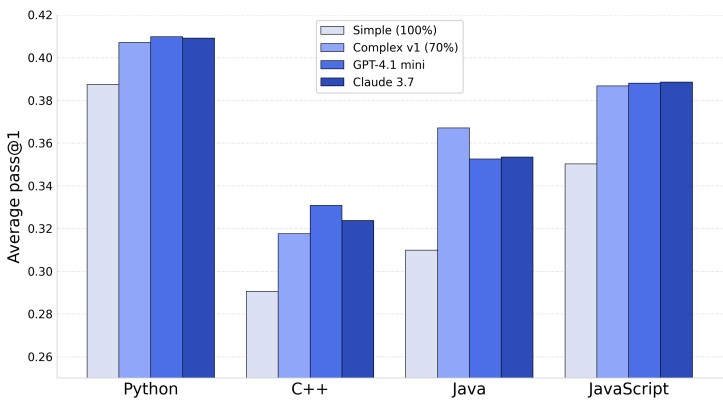

Figure 8: Per language comparison of unit test–based and LLM-based verification strategies.

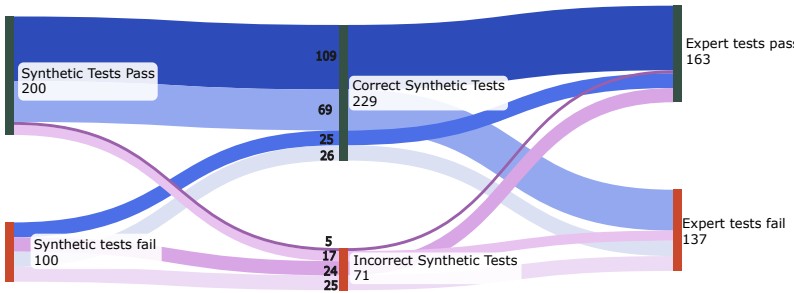

Figure 9: Sankey diagram representing the overlap and distribution of sample counts among two verification processes: Synthetic unit tests verification (pass/fail) and Expert-written unit tests (pass/fail), and the correctness of synthetic unit tests.

## B  ADDITIONAL EXPERIMENTS

### B.1  UNIT TEST SOURCE DIVERSITY

To understand the impact of verifier diversity, we perform a focused ablation comparing training data selected using unit tests generated by a single teacher versus unit tests of equivalent complexity and quantity generated by multiple teacher models. The underlying set of coding problems remains fixed, and candidate code solutions are included based on passing the corresponding unit tests. We find that using multiple unit test generators significantly improves model performance in C++ and Java (by 3 and 1 points absolute gains respectively), while providing marginal or slightly negative effects in Python and JavaScript (1 and 0.1 degradation respectively). These results suggest that diversity in verification sources can enrich training data for some languages, likely by exposing models to a broader set of edge cases and code patterns.

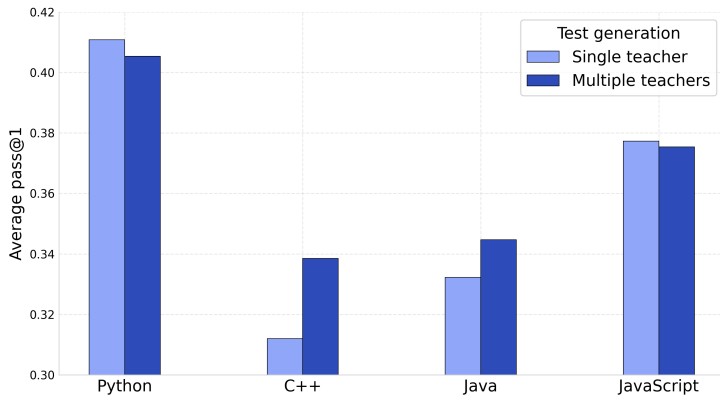

Figure 10: Effect of verifier diversity on model performance. Training data filtered with unit tests from three different generators (vs. a single generator) leads to significant gains in C++ and Java, but shows marginal or slightly negative effects in Python and JavaScript.

### B.2  CODE SOLUTION DIVERSITY

In the paper we examined how test complexity and quantity shape the training distribution by filtering solutions based on correctness and conformity. While these approaches influence which examples are retained, they do not capture a complementary dimension of learning signal: diversity in how a task can be solved. Code generation tasks often include multiple semantically equivalent but structurally distinct solutions. This raises a natural question: *can exposing the model to a broader set of valid implementations per problem improve generalization and pass@1 performance?*

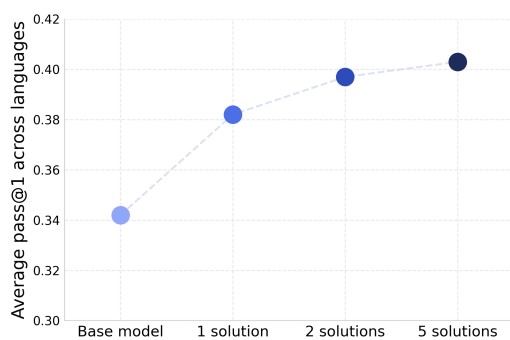

Figure 11: Impact of increasing solution diversity per problem. We compare models trained on 1, 2, or 5 solutions per coding problem while keeping the problem set fixed. Both the 2-solution and 5-solution settings improve average pass@1 across evaluation sets, demonstrating that exposing the model to multiple semantically correct but structurally distinct implementations enhances generalization and downstream performance.

To investigate this, we study the effect of training with multiple solutions per problem. Starting from a 70K subset of our dataset, we generate additional solutions for each problem from a pool of teachers, including Command-A and DeepSeek-V3, Qwen2.5-coder-32B, and llama-3.1 at various temperatures. The pass rate is on average $\tau = 0.5$ for the samples selected. Note that in each experimental variant, the set of code problems remains fixed, while the code solutions vary, allowing us to introduce diversity on the solution side.

As shown in Figure 11, both the 2-solution and 5-solution variants yield improvements in average pass@1 across all evaluation sets. This suggests that, in addition to selecting the right kind of examples, enriching the dataset with diverse variations of correct solutions provides additional generalization signal, helping the model better capture the multifaceted nature of coding tasks.

### B.3 FORMALLY CORRECT VS. INCORRECT: A CONTROLLED COMPARISON

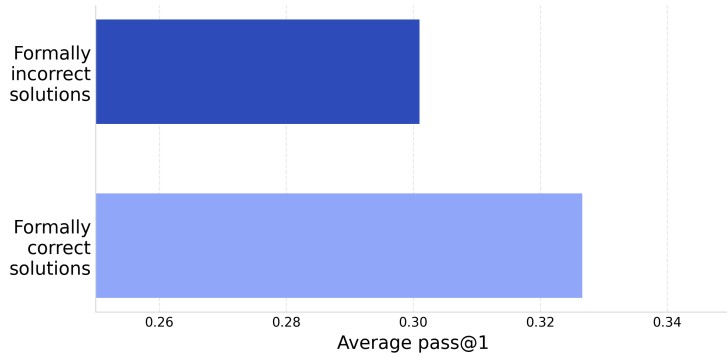

Figure 12: Comparison of models trained on solutions for the exact same problems, where one set passes the tests ("Formally correct") and the other fails ("Formally incorrect"). Models trained on correct solutions outperform those trained on incorrect ones by 3 pass@1 points across all benchmarks, demonstrating that maintaining a verification signal at the solution level is crucial, even when using relaxed thresholds or LLM-based filtering.

Our earlier results suggest that strict verification criteria, such as requiring 100% pass rates can overly constrain the training distribution, excluding useful solutions and reducing downstream performance. However, many of these results involve shifts not only in solution quality but also in problem selection, making it difficult to isolate the impact of verification alone. This raises a natural question: is correctness itself the problem, or is the issue that our definition of correctness is too rigid? To disentangle these effects, we need to isolate the impact of verification from confound-

ing factors like problem selection. To control for this, we design a focused experiment where the problem set is held fixed, and we generate two solutions per prompt: one that passes the test suite ("formally correct") and one that fails ("formally incorrect"). This allows us to directly compare the effects of including verified versus unverified solutions while holding the problem distribution constant. We end up with 41K samples where the formally correct solutions have an average test pass rate of $\overline{\tau} = 0.85$, while the formally incorrect solutions average $\overline{\tau} = 0.05$. Since we require both passing and failing solutions for the same code problems, coverage is bounded by the ability of the teacher model to generate such contrastive pairs, which is non-trivial for harder problems.

We observe in Figure 12 that models trained on the formally correct solutions outperform those trained on the formally incorrect ones by 3 absolute points across all benchmarks. This result reinforces the importance of maintaining some verification signal in the data pipeline: even though relaxing verification thresholds or using LLM-based filters can lead to better results than rigid formal correctness checks, the presence of quality, at least at the solution level, remains crucial. The benefits of relaxing or rethinking verification mechanisms are not due to correctness being irrelevant, but rather to the limitations of overly narrow or brittle correctness definitions (e.g., strict or inaccurate test-based criteria). When solution quality is held constant, better solutions still yield better models, underscoring that verification should be applied adaptively rather than abandoned.

### B.4 SYNTHETIC CODE IS STILL A PRACTICAL ALTERNATIVE TO HUMAN-WRITTEN CODE

Having established the limitations of verification, an important question remains: how much do these constraints actually hurt in practice compared to the gold standard of human-authored code? Human-written solutions provide richer coverage and more diverse styles, but they are expensive to collect at scale. Synthetic data, by contrast, is abundant but ultimately bounded by the verification ceiling we identified earlier.

To examine this trade-off, we directly compare models finetuned on human-written versus synthetically generated code solutions for the same set of programming problems. As shown in Figure 13, models trained on synthetic data achieve performance competitive with those trained on human data, despite the known limitations of synthetic verification. This suggests that current pipelines already approximate much of the benefit of human data, while also underscoring the need to continue improving verification systems to capture correctness beyond their current ceilings.

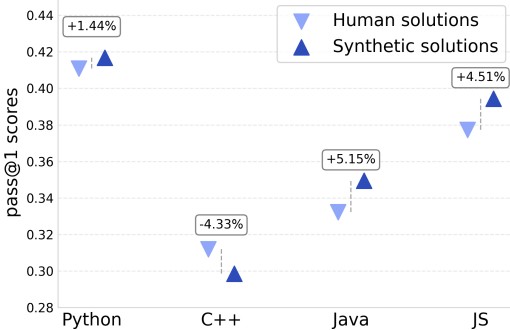

Figure 13: Comparison of models finetuned on human-authored versus synthetically generated solutions for the same set of programming problems. Models trained on synthetic code achieve performance competitive with human-trained models, highlighting that current synthetic pipelines capture much of the value of human data while emphasizing the need for improved verification systems to handle correctness beyond current ceilings.

# C   TEST SUITE GENERATION STRATEGIES

## C.1   MINIMAL TEST GENERATION

The following prompts are used to generate the simple unit-tests in Java, C++, and Python. We ask the model to use some standard libraries for test generation, namely `unittest` for Python, `junit` for Java, and `cassert` for C++.

> USER: Please write some {`language`} code using the {`testing_library`} library to create tests for the following instruction:
>
> {`instruction`}
>
> There should be at least 3 different tests and at least 2 of these tests should be testing inputs that are not trivial. Return only the imports and the test class definition, inside a {`language`} markdown code block.

For C++, we also append the following instruction to the user prompt given above:

> You can assume that the solution code for the problem above is already imported. Please import the {`testing_library`} library, and other libraries that you may need to run the tests. Write your tests in the main function.

For JavaScript, we ask the model to write test code using console.assert as follows:

> USER: Please write some inside a {`language`} code using console.assert(...) to create tests for the following instruction:
>
> {`instruction`}
>
> There should be at least 3 different tests and at least 2 of these tests should be testing inputs that are not trivial. Return only the test code, inside a inside a {`language`} markdown code block. You can assume that the solution code for the problem above is already imported.

## C.2   STRUCTURED TEST GENERATION

The following prompts are used to generate the first version of complex tests.

> USER: Please write some {`language`} code using the {`testing_library`} library to create tests for the following instruction:
>
> {`instruction`}
>
> You are also provided with a code solution and some existing tests corresponding to the instruction given above. Code solution:
>
> {`code`}
>
> Existing unit-tests:
>
> {`test`}
>
> There should be at least 4 tests different from the ones already provided to you and they should target more edge cases, and focus on testing different parts of the code. Aim to generate 8 diverse tests in total while carefully testing for error handling, and at least 4 of these tests should be testing inputs that are not trivial.
> Return only the imports and the test class definition, inside a {`language`} markdown code block.

As with Minimal unit-tests, we append the additional instruction for C++. Similarly, for JavaScript, we prompt the model to generate tests using `console.assert` using the prompt below:

---

USER: Please write some inside a {`language`} code using console.assert(...) to create tests for the following instruction:

{`instruction`}

You are also provided with a code solution and some existing tests corresponding to the instruction given above. Code solution:

{`code`}

Existing unit-tests:

{`test`}

There should be at least 4 tests different from the ones already provided to you and they should target more edge cases, and focus on testing different parts of the code. Aim to generate 8 diverse tests in total while carefully testing for error handling, and at least 4 of these tests should be testing inputs that are not trivial.
Return only the imports and the test class definition, inside a {`language`} markdown code block. You can assume that the solution code for the problem above is already imported.

---

## C.3    CONTRASTIVE TEST GENERATION

The following prompts are used to generate the more complex contrastive set of test suites.

---

USER: Please write some {`language`} code using the {`testing_library`} library to create tests for the following instruction:

{`instruction`}

You are also provided with a few candidate code solutions corresponding to the instruction given above. Candidate Code solutions:

{`code_solutions`}

Target your test generation such that at least one of them fails for each of the solutions provided above.
Also, carefully take care of the following while generating tests:
- The tests should focus more on testing edge cases (eg. inputs that are rare)
- The tests should ensure coverage of all output categories. Eg. for a problem with a binary yes/no answer, the tests should focus on inputs that test both outputs equally.
- The test should cover a diverse range of different input types also testing for invalid input types.
- While basic inputs should be covered, the tests should be aimed at harder inputs as well.
- Also try to include larger/complex inputs that'd fail a correct but un-optimized code solution.
On average, aim to generate 5-8 tests in total.
Return only the imports and the test class definition, inside a {`language`} markdown code block.

---

As with Minimal and Structured test suites, we prompt the model to use `console.assert` for test generation in JavaScript.

# D    DIRECT LLM VERIFICATION PROMPT

USER: Your goal is to assess the correctness of a given code solution corresponding to a given code problem. Your assessment should be thorough and based on the following criteria: - Does the code correctly implement a solution to the given code problem?
- Is the code correct for all types of edge cases (eg. inputs that are rare and require special conditions etc.)
- Does the code ensure correct coverage of all output categories. eg. for a problem with a binary yes/no answer, does the code correctly cater to both categories?
- Does the code produce correct output for a diverse range of different input types? Does the code implement proper error handling for invalid input types?
- Does the code contain sufficiently optimized logic for harder and more complex input types? Does it do a good job as far as time and space complexity are concerned?
Code problem:

```
{instruction}
```

Code Solution:

```
{code_solutions}
```

OUTPUT FORMAT: Your output should be a JSON with the following two keys: a binary 0 or 1 score depending on whether the code is correct or not and describing your analysis of the score.

```
{
"assessment_explanation": "<explanation>",
"score": <0/1>",
}
```

# E    LLM USAGE DISCLOSURE

We made use of LLMs during the preparation of this paper. Specifically, LLMs were employed to polish the writing for clarity and consistency and assist in polishing the plots. All substantive research ideation, contributions, system development, and analysis were carried out by the authors.

