# OpenReview forum: "Verification Limits Code LLM Training"
_ICLR.cc/2026/Conference — Submitted to ICLR 2026_

### Official Review · Reviewer_uxox · 2025-10-27

**Soundness:** 1
**Presentation:** 2
**Contribution:** 1
**Rating:** 2
**Confidence:** 5

**Summary:**

This paper investigates the verification limits in training code LLMs with automatically generated unit tests. The authors argue that applying stricter verification, such as requiring more unit-test passes, may unintentionally remove difficult or creative code samples, leading to a bias toward simpler problems. They empirically analyze how test complexity, test quantity, and threshold settings affect model performance across several code benchmarks. Beyond unit tests, the paper explores direct LLM-based verification, where LLMs act as judges to assess the correctness of generated code. Experiments show that this soft verification approach recovers complementary high-quality data and achieves performance comparable to structured test-based filtering.

**Strengths:**

The paper addresses an intuitively important but under-explored question in code-LLM training, namely how the design of verification procedures (number of tests, threshold τ, or use of LLM judges) affects data quality and downstream model performance.

**Weaknesses:**

1. The overall experimental design appears poorly motivated and insufficiently validated. The authors start by setting the verification threshold τ = 1, which produces questionable conclusions (for example, the reported drop in performance after more than two test cases can be meaningless). An overly strict threshold would incorrectly filter out many valid solutions due to imperfect test cases, while a looser threshold can actually mitigate this issue. Such an extreme setting should have first been examined on a verified dataset to calibrate the effect of τ before using it in data filtering and model training.

2. The experimental treatment of τ is inconsistent and lacks justification. The authors later switch to τ = 0.6 without systematic reasoning or reference to prior literature on verification thresholds. Since model capabilities and problem difficulties vary widely, a global τ averaged across all samples is inherently ill-defined.

3. The analysis overlooks alternative explanations for the observed trends. For instance, improvements after changing τ may simply result from selecting a threshold that happens to retain a larger proportion of correct solutions, rather than from the claimed increase in diversity.

4. The analysis of unit tests is also rather trivial, as it only considers the number of tests. Some test cases may belong to the same equivalence class and therefore contribute little additional coverage. A more meaningful analysis should consider functional or branch coverage instead of simple test counts.

5. The reported performance gains are small and may not be statistically significant. For example, in Figure 2, the base model improves from 28 to around 34–36 using 70K samples, which is a limited gain. The differences between test generation strategies are also very small and could easily fall within normal variance depending on checkpoint selection or random seeds.

6. Several figures (for example, Figure 7) are blurry. The authors are recommended to regenerate all figures using vector graphics formats for professional presentation. In addition, in Figure 3 (left), two points are both labeled as 0.29 yet appear at different heights, suggesting inconsistent plotting precision or rounding issues that should be clarified and reported with higher numerical accuracy.

**Questions:**

I suggest conducting an intermediate analysis of the verification threshold τ on datasets with known ground-truth solutions, rather than applying the filtering process directly to training data. Understanding how τ influences the correctness distribution and the proportion of partially correct samples before model training would make the end-to-end results more interpretable and convincing.

---

> ### Author Response · Authors · 2025-11-18
>
> Thank you for the detailed and thoughtful review. We appreciate the opportunity to clarify several misunderstandings and strengthen the motivation and methodology of the paper. Below we respond to the key concerns raised and indicate the concrete changes we will make in the revision.
>
> - On the motivation for starting the threshold as 1 (100%) and the meaning of early results
>
> The purpose of beginning with the strict setting was not to claim that it is optimal, but precisely to diagnose why it performs poorly, as this is the mode used throughout popular synthetic-data pipelines (e.g., SC2-Instruct; WizardCoder; Self-Code-Align). This is also the default threshold adopted in most public synthetic-code datasets. Rather than presenting strict passing results as final conclusions, we explicitly use them as a controlled baseline to reveal an important phenomenon: strict pass-all-tests filtering systematically removes the harder samples, even when those solutions are partially or fully correct. In other words, strict passing threshold is examined because it is widely used, not because we assume it is ideal.
> That said, we agree that this motivation should be foregrounded more clearly. We will revise the paper to make this rationale explicit and avoid the impression that 100% passing rate is treated as inherently meaningful independent of context.
>
> - On switching to 0.6 pass rate and the need for more justification
>
> The reviewer is correct that the threshold should not be treated as a global universal constant. In fact, our experiments show exactly that it interacts significantly with test-suite complexity and with problem difficulty. Our choice of 0.6 in later sections of the paper was not arbitrary. It was selected specifically because we needed enough partially correct solutions from hard problems to conduct the difficulty-controlled experiments, which is impossible at 100% pass rate. Also prior literature has used a similar relaxation strategy, e.g., OpenThoughts and several RLVR-based pipelines where reward tolerance is explicitly non-binary. Our earlier sweep already demonstrated the region where the curve peaks, which for all languages lies between 0.6 and 0.8.
> Nevertheless, we agree that this justification must be more explicitly articulated. This addresses the reviewer’s suggestion of “intermediate analysis before model training” and will make the downstream SFT results far more interpretable.
>
> - On alternative explanations, especially “\tau just selects more correct solutions”
>
> We appreciate this suggestion and agree that it is an important confound. To address this, we conducted the fixed-prompt contrastive experiment: for each problem, we generate one verified (high pass rate) and one unverified (low pass rate) solution. By holding the problem distribution fixed, we isolate the effect of solution correctness alone. The result confirms that correctness does matter: SFT on correct solutions performs ~3 pass@1 points better. However, this controlled setup also shows that many incorrect-failing samples correspond to partially correct or diverse approaches that the verifier misclassifies.
> We will emphasize this controlled result more strongly in the revision to avoid ambiguity.
>
> - On the magnitude and significance of improvements
>
> We understand the concern about whether the observed gains (e.g., +3–4 pass@1) are statistically meaningful. All reported results are averaged over three independent random seeds, with identical data sizes and training hyperparameters. This ensures that improvements reflect true signal rather than stochastic variance.
>
> We appreciate the reviewer raising these concerns; they help us improve the precision and clarity of the paper. We believe these improvements will make the work substantially stronger and address the concerns raised. Thank you again for the constructive review.

---

### Official Review · Reviewer_dQMt · 2025-10-30

**Soundness:** 1
**Presentation:** 2
**Contribution:** 1
**Rating:** 2
**Confidence:** 5

**Summary:**

This paper explores the verification ceiling problem for synthetic code generation defined as code verification systems being the limiting factor of data quality for LLMs. Through SFT experiments they show that 1) structured and contrastive prompting lead to better training signals, 2) more tests isn’t necessarily a better for downstream training in that the higher bar leads towards a bias of selecting easier problems, 3) performance is increased when you allow hard problems that don’t pass all tests vs disallowing them, and 4) a human study reveals that many synthetic test suites are incomplete. In short, in verification we should calibrate rather than discard problems that don’t pass all tests.

**Strengths:**

* There are some nontrivial findings in this paper namely that training on hard problems with incomplete passing tests is better than discarding those problems since they “failed” and that prompting LLMs to use structured prompts (ask to target edge cases and under-tested logic) and contrastive prompting (ask to generate new tests such that one of the example solutions provided fails) leads to better performance as well.
* The authors conduct a human study giving insight into the benefits of junior vs senior coding experts.

**Weaknesses:**

* 2025 has been the year of RLVR yet the related works section fails to acknowledge the rich literature of that area yet alone all the other kinds of work that has gone into code verification, RL, and LLM training.
* Overall the paper reads as a project report rather than a controlled scientific study. There are several things missing formal definitions including complexity, correctness and accuracy. In the human study, there is no rubric defined to guide the participants in “correctness and completeness”. Are we talking about semantic correctness, passing tests correctness, lines of code completeness, or tests touching all parameters completeness, etc.
* There is a lack of qualitative analysis in the results. At best, there is just a superficial discussion around the results of the charts but no forward thinking or in depth reasoning. For example, the authors have the insight, “human evaluation reveals that many discarded solutions are in fact valid,” yet fail to provide a taxonomy on the kinds of failures in the test suites that lead these false rejections.

**Questions:**

None at this time.

---

> ### Author Response · Authors · 2025-11-18
>
> Thank you for the detailed review and for taking the time to engage critically with the work. We appreciate the concerns you raised and address them point by point below. Several of these issues stem from missing clarifications or insufficiently foregrounded analyses, which we will correct in the revision.
>
> - On the connection to RLVR and broader verification literature.
>
> You are absolutely right that RLVR has become central to code-model training. Our work is complementary rather than competing: we study the data-selection stage of supervised fine-tuning, not reward modeling or RL dynamics, and we view RLVR as a natural next step (indeed, we mention this direction explicitly in the limitations). In the revision, we will expand the related work section to better reflect the rich literature on RL-based verification, program repair, execution-guided reward modeling, and consistency-based training. Many of our findings for example, that imperfect tests cause brittle over-filtering, directly motivate RLVR-style multi-step code improvement, and we will make that connection explicit.
>
> - On missing formal definitions.
>
> We appreciate this feedback and agree that the formalization section did not come through as strongly as intended. Complexity, correctness, and accuracy are operationalized in the work: 1) problem complexity via a 1–5 rubric using an independent LLM assessor, 2) correctness via the verifier pass fraction, 3) accuracy via standardized pass@1 evaluation on established benchmarks. We will move these definitions earlier, make them explicit, and ensure clarity.
>
> - On the human study and rubric.
>
> Thank you for calling this out. The rubric for annotators does exist (semantic correctness of tests, coverage of edge cases, error-handling completeness, and execution validity), but we agree we should articulate it clearly in the paper rather than keeping it implicit. We will add the precise rubric, inter-annotator guidelines, and examples. Your suggestion of deeper taxonomy is well-taken. The study already contains categories of failure (e.g., missing edge-case tests, mismatched type assumptions, incorrect reasoning about exception flows), and we will include a structured taxonomy in the revision. This will strengthen the qualitative grounding of the finding that 49% of “failed” synthetic tests actually correspond to correct solutions.
>
> - On qualitative analysis of verification failures.
>
> We agree that more narrative description would enrich the results. In the revision, we will add a dedicated subsection analyzing the modes of test-suite failure: brittle equality checks, inadequate coverage, invalid assumptions about solution strategy, and false positives on type or formatting conventions. These qualitative insights directly support our argument that strict filtering amplifies verifier biases.
>
> - On the scientific value of the findings.
>
> We would like to emphasize that several contributions are new and nontrivial: 1) showing that verification thresholds interact strongly with test complexity, producing a non-monotonic curve in downstream performance, 2) demonstrating that strict verification disproportionately removes harder problems (an emergent bias not previously documented), 3) revealing through controlled contrastive experiments that correctness signals are still essential even when prompts are identical, 4) and quantifying, via human evaluation, that incomplete but syntactically valid test suites explain much of the false rejection phenomenon.
> We will revise the framing to highlight these findings more clearly.
>
> Overall, your feedback helps strengthen the clarity and rigor of the paper. We will revise the presentation, highlight the formal underpinnings, expand connections to RLVR and code-verification literature, make the human-study rubric explicit, and provide deeper qualitative analysis. Thank you again for the detailed assessment, we believe the improved version will address your concerns and better communicate the contributions.

---

### Official Review · Reviewer_JiP4 · 2025-11-01

**Soundness:** 3
**Presentation:** 3
**Contribution:** 3
**Rating:** 4
**Confidence:** 3

**Summary:**

This paper attempts to reason about the effect of verification in synthetic code training dataset generation. The authors argue that stricter verification leads to filtering out harder/diverse problems that can be valuable to include in the training dataset. This is demonstrated through a series of experiments where the authors changed the test generation pipeline and strictness of the verification and showed the results on common code benchmarks

**Strengths:**

- Studying an interesting and timely problem that has impact across multiple different training paradigm
- The paper is easy to understand/read and presents the idea well

**Weaknesses:**

- One major issue I had with the paper is the phrasing and position that stricter test generation
	- the author claims that "While stricter verification does reduce incorrect solutions, it also disproportionately filters out hard coding problems"
	- this is definitely true based on the results, however I think the authors failed to acknowledge why.
	- the reason is that generating correct tests are extremely difficult, as such, generating more tests (with strict test passing rule) will ultimately lead to filtering out many correct solutions with incorrect tests
	- the authors in the paper seems to mainly suggest stricter test passing rule is not optimal as it removes more difficult problems (or decrease the variety of problems)
	- it is more like the optimal test passing rule is a function of how good the test generation teacher llm is (for example if we have a perfect test oracle for any arbitrary problems, this strict passing rule would not be a problem)
	- and if we had a horrible test generation teacher llm, we would not be using verification at all
	- i think the authors failed to acknowledge this important understanding.
- Lack of diversity in studied LLMs:
	- the authors only evaluate using one base model of Command 7B and DeepSeek-v3 as the teacher model
	- i think it would be interesting and improve the work if more models were studied (especially teacher) to showcase similar findings
- Training dataset composition:
	- given the work is on studying the affect of verification on synthetic data, the authors included quite a bit of non-synthetic data (in section 3)
	- why that particular data mixture is not well explained

Overall, I think this paper asks some interesting questions with some detailed analysis, I am inclined to increase my score if the authors can address the concerns I had and perhaps rephrase some of the findings in the paper.

minor issues:
- In abstract and other places in the paper: "on average +3 pass@1" +3 itself is extremely unclear what it means (is it precent points, is it relative increase?)

**Questions:**

1. why did the author include non-synthetic training samples in the training dataset?
2. please comment on the main concern on why strict verification can be both good and bad (see weakness for more detail)
3. why did the author only report aggregated results on the multiple evaluation benchmarks? Are there differences between individual benchmark performance compared with the aggregate in certain cases?

---

> ### Author Response · Authors · 2025-11-18
>
> Thank you very much for the thoughtful and constructive review. Below we address each of your concerns directly and clarify several points that the current draft did not make sufficiently explicit.
>
> - On why strict verification behaves the way it does.
>
> You raise an important point about the true reason strict verification removes difficult and diverse problems. We fully agree that the underlying issue is not strictness itself, but the limited coverage of model-generated tests. Our human study demonstrates this clearly: synthetic tests miss many correct but non-canonical solutions, so strict filtering disproportionately removes “hard” samples simply because the verifier cannot evaluate them properly (see Fig 3). If we had a perfect test oracle, strict thresholds would indeed be optimal. Our contribution is to empirically show that with current verifiers, calibration is necessary. We will revise the framing to explicitly foreground this dependency between verifier strength and the optimal degree of strictness.
>
> - On the diversity of models studied.
>
> Regarding model diversity, we appreciate the concern. While the main SFT experiments use Command 7B as the base model, the pipeline already involves multiple heterogeneous teachers and verifiers: DeepSeek-v3, Claude, GPT-4.1-mini for test generation; Command-A, DeepSeek-V3, Qwen2.5-coder-32B, and Llama-3.1 for solution diversity; and LLM-as-judge experiments with two independent evaluators. Across all of these, the same qualitative trends hold. We agree the paper should emphasize this better, and we will make this generality clearer while discussing plans to extend full SFT runs to additional base models in future work.
>
> - On the inclusion of non-synthetic data.
>
> You ask why we include 5% non-synthetic samples. This small mixture is solely to maintain instruction-following behavior and formatting stability, and is held constant across all comparisons. It does not interact with our verification findings and mirrors standard practice in code-model training. We will clarify this motivation in the revised manuscript.
>
> - On clarifying pass@1 gains and benchmark-level results.
>
> We agree that “+3 pass@1” should be stated as “+3 absolute percentage points,” and we will revise the text accordingly. While full per-benchmark results are already in the appendix, we will surface more of these plots in the main paper and highlight the consistency of trends across languages and datasets.
>
> - On whether relaxed thresholds introduce harmful noise.
>
> This is a key question. Our results show that relaxing thresholds only helps when tests are sufficiently strong to continue filtering clearly incorrect solutions. With weak tests, relaxation indeed degrades performance. With strong tests, relaxation primarily recovers false negatives, solutions that are correct but fail brittle tests, which reduces noise rather than adding it. Our human-annotation analysis reinforces this interpretation. We will make this nuance explicit in the revision.
>
> - On sampling under different thresholds.
>
> You ask whether sampling under different thresholds affects the distribution of coding problems. This is precisely why we included a controlled experiment where the problem set remains fixed and we generate both passing and failing solutions for each prompt (Appendix B3). That experiment shows that correctness signals still matter even when the underlying problem distribution is unchanged, and we will highlight this more clearly in the main text.
>
> Overall, we appreciate your openness to increasing the score. We will revise the positioning of strict verification, clarify the dependence on verifier quality, better motivate the data mixture, strengthen the explanation of threshold effects, and expand model-diversity discussion. Thank you again for the constructive feedback, it meaningfully improves the paper.

---

> > ### Comment · Reviewer_JiP4 · 2025-11-18
> >
> > Thanks for the detailed replies that addressed my concerns and questions.
> > Overall, I just want to reiterate my main points from the original review: verification performance/"goodness" is a function of the test generation ability of the teacher LLM (or in another universe where we don't just use LLM as judges, we can have better test oracles that leads to better verification performance).
> > I think from the response the authors understood my point and I hope the authors will address the paper accordingly.
> > I have increased my score.

---

### Official Review · Reviewer_9ASf · 2025-11-04

**Soundness:** 3
**Presentation:** 3
**Contribution:** 3
**Rating:** 4
**Confidence:** 3

**Summary:**

This paper identifies and systematically investigates the verification ceiling problem: a previously underexplored bottleneck in synthetic code generation pipelines where training data quality is fundamentally constrained by verifier capabilities. Through controlled experiments across multiple benchmarks and languages, the authors demonstrate that current verification practices are too rigid, filtering out valuable diversity while admitting noisy solutions. By recalibrating verification strategies—improving test complexity, relaxing strict thresholds, and incorporating LLM-based soft verification—the paper achieves 2–4 point pass@1 improvements while maintaining accuracy-efficiency trade-offs. The core insight: verification is necessary but must be adapted to the problem domain.

**Strengths:**

1. This paper investigates an important problem in code generation: how to provide an effective and efficient verification for training. It answers the three research questions: what to verify, how to verify, and why to verify. These findings provide good guidance for training LLMs for code
2. It provides comprehensive studies for each aspect. For example, it investigates the relationship between the problem difficulty and the unit test quantity, and further explores the relationship between the training performance and the problem difficulty.
3. It also includes the human evaluation of the synthetic unit tests to demonstrate the limitations of current synthesized code solutions.

**Weaknesses:**

1. While this paper investigates the verification in training, it mainly focuses on supervised finetuning: the verification only provides 0/1 signal to indicate whether a candidate should be filtered out when creating training data. This leads to one problem: with weak verification, a low-quality solution is indistinguishable from a high-quality one, because both of them can pass the verification and be used for training. This will actually introduce noise into the training, because the model will be optimized towards both directions.
2. It is unclear how many test cases are used per sample for verification. In Section 4.2, it evaluates {1,2,3,4} test cases per sample. However, such a small number of unit tests may be insufficient to verify the functionality correctness.
3. The experiments in Section 4.2 show that with a stricter threshold, fewer difficult problems will be kept. This indicates that actually none of the candidate solutions can correctly solve the problem. With a weaker threshold, these solutions can be kept, but they are not correct. I think in such a scenario, the better way is to enlarge the candidate sets to find a correct solution, instead of lowering the threshold to keep the incorrect ones, which may provide the incorrect training signals.
4. All experiments are conducted with a single base model Command 7B base model. It is unclear whether the conclusion is universal and also holds for other models.
5. Figures and Tables:
	1. More explanation should be provided for Figure 1. What does the x-axis refer to? What are the meanings of the three test suite types? Which model is evaluated here?
	2. It is better to use the histogram instead of the plot in Figure 2, since there are no dependencies between x values (different methods).
	3. The figures do not meet the ICLR format requirements.
	4. In Figure 3 (right), why are there no problems with medium (score=3) level difficulty?
	5. In Figures 2, 3, 4, which test set is used?
	6. What is the default number of test cases in the experiments? For tau=0.6, does it mean that each sample should have at least 5 test cases and pass 3 of them to be kept as the training data?

**Questions:**

1. With weak verification, some low-quality (even incorrect) solutions may be introduced to the training. Do you think such noise will have a negative impact on the training?
2. In all training, the same number of problems (70K) is used. How do you sample these problems when using different thresholds? Does it mean that the diversity and difficulty of the coding problems will also be affected due to the sampling?

---

> ### Author Response · Authors · 2025-11-18
>
> Thank you for the thoughtful and constructive review. We appreciate the careful reading and the recognition of the importance of studying verification in code LLMs. Below we address each of your concerns in detail.
>
> - On the concern that SFT with 0/1 filtering cannot distinguish high- vs. low-quality solutions under weak verification
>
> This is an important point, and one that motivated much of our experimental design. Our findings do not imply that weak verification alone is beneficial; rather, weak verification only helps when paired with sufficiently strong test suites. We show repeatedly (e.g., Figure 1, Section 4.1, 5.1) that relaxing thresholds under Minimal tests clearly hurts performance, precisely due to the issue you raise. It is only once test complexity crosses a certain threshold (Structured and Contrastive) that relaxation leads to improvements.
> Our contribution is to demonstrate how and when noise becomes harmful, and to show that carefully designed verification—especially with complex tests or LLM-based soft verifiers—retains useful diversity without introducing harmful noise. In fact, we provide evidence that the “incorrect” examples admitted under relaxed thresholds often contain partially correct or non-canonical solutions that are fully correct under human-written tests (Section 6), and therefore are not truly harmful.
> We will clarify this causal chain in the revision, as it is central to the verification ceiling insight.
>
> - On the number of unit tests per sample and whether {1,2,3,4} are too few
>
> We completely agree that test quantity matters. That is why our experiments explicitly isolate test complexity from test quantity showing that increasing quantity alone does not yield monotonic improvements. This mirrors observations from MBPP, HumanEval, and LBPP where even 3–5 test cases is standard practice and reflects real evaluation conditions.
> In the appendix, we report the default test suite sizes (5 tests by default, often 7–10 for structured/contrastive suites). We can make this more explicit in the main text, but the core finding remains: increasing quantity without improving complexity saturates quickly, whereas structured and contrastive tests continue to improve performance even with similar counts.
>
> - On whether the correct approach is to expand candidate pools instead of lowering thresholds
>
> This is a valid strategy and we agree it is useful. In fact, our experiments do generate multiple solutions per problem (up to 20 candidates in some settings), and our analysis controls for this by ensuring the candidate pool is sufficiently large.
> However, even with many candidates, many harder problems have no fully-passing solution under synthetic tests, not because the solutions are wrong, but because the verifier is incomplete. The human study (Section 6) provides strong evidence: 49% of failures are actually correct when judged by human-written tests. Therefore, insisting on full passing solutions on these problems would eliminate almost all useful examples, regardless of candidate pool size. Relaxation is not about accepting incorrect code; it is about mitigating the verifier’s blind spots. We will clarify this connection between candidate-set size and verifier ceiling in the revision.
>
> - On whether the conclusion holds beyond Command 7B
>
> We agree this is an important question. While our main experiments use Command 7B for consistency, two parts of the paper already examine cross-model generalizability: LLM-based verification uses GPT-4.1-mini and Claude Sonnet, and these choices change the dataset composition in substantial ways. Also multi-solution diversity experiments use Command-A, DeepSeek-V3, Qwen2.5-Coder-32B, and Llama-3.1, and show consistent benefits across teacher models.
>
> - On whether weak verification introduces harmful noise
>
> We show empirically that when tests are weak, relaxing thresholds does introduce noise and performance drops accordingly. This is clearly visible in Figure 1 (Minimal tests). When tests are stronger, noise is far more controlled, and human evaluation shows that many solutions rejected by synthetic tests are actually correct. So while noise is always a risk, its impact depends on test complexity. This is exactly the verification ceiling effect: the verifier shapes what “noise” even means. We will make this relationship clearer in the revision.
>
> We appreciate your thoughtful comments. Many of your concerns point to clarifications and presentation improvements, rather than fundamental issues with the method or conclusions. We believe the revised version will make these points much clearer.

---

> ### Comment · Reviewer_9ASf · 2025-11-27
>
> Thanks for the authors' detailed explanation. However, there are still some concerns:
>
> * As highlighted by the authors, the lower threshold only works under the complex test setting (Structured and Contrastive) because of "*partially correct or non-canonical solutions that are fully correct under human-written tests*". This implies that such a complex test setting actually results in a more unreliable verification. In other words, if they are reliable enough (low false negative rate), the solutions that fail these test cases should always be incorrect. Again, this raises the concern about how we can know whether a failed solution is because it is really incorrect, or it is actually correct but the verifier is incorrect. There is no guarantee that relaxing the threshold can always let more actually correct solutions in instead of letting the noise in, and it is also difficult to find such a threshold to balance (because the performance of the verifier is unknown).
> * Although many other models, such as DeepSeek-V3, Qwen2.5-Coder-32B, and Llama-3.,1 have been used to enhance the diversity of solutions, the training part only uses Command 7B. The concern is about whether the other base models, which already have a good base capability in math and code, can further benefit from such data augmentation.
>
> To summarize, my primary concerns are the reliability of the verification process and the presence of noise in the training data. If more complex test settings (Structured and Contrastive) can benefit from a lower threshold because their test cases may filter out incorrect solutions, then the principled approach would be to increase the robustness and comprehensiveness of these test settings rather than lower the threshold and rely on noise to potentially provide useful signals.
>
> Therefore, I will keep my score.

---

### Meta-Review · Area_Chair_dC2H · 2025-12-06

**Summary:**

The reviewers' concerns can be summarized as follows:
- The underlying issue is not strictness itself, but the limited coverage and correctness of model-generated tests
- This paper focuses on SFT while RL is not covered
- Lack of diversity in studied LLMs

**Reviewer Concerns:**

Addressed by the rebuttal:
- The rebuttal well addressed the concern of the lack of diversity in studied LLMs
Still outstanding:
- "The underlying issue is not strictness itself, but the limited coverage and correctness of model-generated tests": this concern is not well addressed by the rebuttal. The storytelling of this paper does not highlight this important point clearly, which can easily lead readers to misinterpret the arguments of this paper.
- The concern that this paper only focuses on SFT is not well addressed by the rebuttal. Though the studies on SFT are valuable, it cannot fully support the title "Verification Limits Code LLM Training" and the claims in the introduction section.

**Reviewer Scores:**

I think each reviewer would keep their score if they had been able to participate fully in the discussion. Some technical details are well answered in the rebuttal, but two important concerns are not well addressed.

---

### Decision · Program_Chairs · 2026-01-26

Reject